# Cutaneous Plasmacytoma with Systemic Metastases in a Cape Serotine Bat (*Laephotis capensis*)

**DOI:** 10.3390/vetsci11020072

**Published:** 2024-02-05

**Authors:** Louise van der Weyden, Alida Avenant, Nicolize O’Dell

**Affiliations:** 1Wellcome Sanger Institute, Wellcome Genome Campus, Cambridge CB10 1SA, UK; lvdw@sanger.ac.uk; 2Department of Paraclinical Sciences, Faculty of Veterinary Science, University of Pretoria, Onderstepoort 0110, South Africa; alida.avenant@westerncape.gov.za; 3Centre for Veterinary Wildlife Studies, Faculty of Veterinary Sciences, University of Pretoria, Onderstepoort 0110, South Africa

**Keywords:** tumour, lymphocyte, B-cell, plasma cell tumour, plasmacytoma, MUM-1, metastasis, bat

## Abstract

**Simple Summary:**

Although they have relatively long life-spans, there have been very few reports of cancer in bats. In this report, we describe a bat that had been caught by a domestic cat and presented with a skin mass over the chest area. Sampling of tissue from the mass revealed sheets of round tumour cells, occasionally appearing to form packets. The nuclei of the tumour cells showed great variability in size and shape with a large proportion of the tumour cells undergoing mitosis. Immunostaining showed positive labelling for MUM1 in the tumour cells. The diagnosis was extramedullary plasmacytoma (EMP); a neoplasm consisting of plasma cells derived from B lymphocytes. Due to a worsening condition two weeks later, the bat was anaesthetised, and the mass surgically removed. Sampling of tissue from the excised mass showed the same tumour cell population as initially observed; in addition, there was local spread of the tumour cells into the spleen and a mild presence of neoplastic cells in circulation. The diagnosis was EMP with systemic metastases. This is the first report of an EMP in a bat, and we compare the findings with that seen in dogs and cats.

**Abstract:**

Despite their relatively long life-spans, reports of neoplasia in bats are rare and are limited to a handful of cases. In this report, we describe a 2-year-old female wild Cape serotine bat (*Laephotis capensis*) that had been caught by a domestic cat and presented with a skin mass over the chest area. Histopathological analysis of a subsequent biopsy revealed proliferating sheets of neoplastic round cells, occasionally appearing to form packets, supported by a fine, fibrovascular stroma. Marked nuclear pleomorphism was seen, as well as a high mitotic count. Immunohistochemistry displayed positive labelling for MUM1 in the neoplastic cells. The diagnosis was extramedullary plasmacytoma (EMP); a neoplasm consisting of plasma cells derived from B lymphocytes. Due to a deteriorating condition, the bat was anaesthetised, and the mass was surgically removed two weeks later. However, the bat succumbed under the anaesthetic. Histopathological examination of the mass showed the same neoplastic cell population as observed in the biopsy; in addition, there was a locally extensive infiltration of neoplastic cells in the spleen and a mild presence of neoplastic cells in circulation. This is the first report of an EMP in a bat, and we compare the findings with that seen in dogs and cats.

## 1. Introduction

Plasmacytomas (also known as plasma cell tumours) are neoplasms consisting of plasma cells derived from B lymphocytes and are divided into two types: solitary plasmacytoma of the bone (SPB) and extramedullary plasmacytoma (EMP, which affects the soft tissue). In animals, plasmacytomas most commonly arise in dogs and comprise ~2.5% of all canine neoplasms, tending to occur mainly in older animals, with some breeds over-represented [1,2,3,4,5,6]. One study found the most frequent site of origin to be the skin [5], whilst other reports suggested oral plasmacytomas are the most common [1,2,3]. The incidence of plasmacytomas in cats is much lower, and the literature mostly consists of case reports of a single animal [7,8,9,10,11], although there are some studies with larger sample numbers (*n* = 7–11) [12,13,14,15].

In dogs, most cutaneous EMPs are benign solitary nodules, and complete excision is usually curative, although recurrence has been reported in some cases [3,6]. Indeed, although there have been reports of metastasis in a few cases of canine EMPs, these were located in the gastrointestinal tract and oral cavity, thus not of cutaneous origin [16,17,18,19]. By contrast, one study of cutaneous EMPs in felines found 4/11 (36%) cases developed metastasis [14], and metastasis has also been reported in EMPs from other sites [8].

In bats, there are very few reports of tumours [20,21,22,23,24,25], and this is the first report of a tumour in a Cape serotine bat (*Laephotis capensis*) and the first report of a plasmacytoma in bats. In this study, we present the details of the case, discuss the differential diagnoses, and compare the findings with that seen in canine and feline EMPs.

## 2. Case Presentation

An approximately 2-year-old female wild Cape serotine bat (*Laephotis capensis*) that had been caught by a domestic cat in Velddrif, Western Cape, South Africa, was presented to the veterinarian for medical care associated with the cat bite. At the time of examination, a mass (~1.5 cm^2^) was noted on the skin over the left lateral chest area. The bat was treated symptomatically and was placed on antibiotic therapy prophylactically for the cat bite and taken to a bat rehabilitation centre in Gauteng (Think Bat; https://thinkbat.org/, accessed on 15 December 2023 ). Despite the weeklong treatment, the mass had not decreased in size. A fine-needle aspiration was performed on the mass, and the cytology smears showed the presence of small neoplastic round cells. To aid the diagnosis, an incisional biopsy of the mass was subsequently performed and submitted to the Histopathology Laboratory of the Faculty of Veterinary Science, University of Pretoria for routine histopathology. Sections (4 μm) were stained with haematoxylin and eosin (HE).

Examination of the HE sections revealed the entire tissue sample consisted of a proliferation of neoplastic cells with a pseudocapsule visible at one edge, and no overlying skin or adnexal structures were present. Extending to all of the tissue margins were proliferating sheets of round neoplastic cells, occasionally appearing to form packets, supported by a fine, fibrovascular stroma (Figure 1a). The neoplastic cells had clear cell borders and scant-to-moderate pale eosinophilic cytoplasm, often with a perinuclear clear zone visible (Figure 1b). Marked nuclear pleomorphism was seen, with nuclei varying from round to oval to indented and occasional bi- and multinucleation. Chromatin was finely stippled and had large central magenta nucleoli. The mitotic count was high, averaging 71 mitoses over 10 high-power fields (Figure 1c). Scattered smaller lymphocytes (confirmed with CD3 and CD20 immunohistochemistry (IHC)) were present between the neoplastic cell population. These findings were highly suspicious for an EMP, and it was decided to confirm this using IHC to evaluate the expression of multiple myeloma oncogene-1 (MUM1) using a monoclonal mouse anti-MUM1 antibody (clone Mum-1p, DakoCytomation, Glostrup, Denmark, code M7259). After dewaxing, sections were treated with 3% hydrogen peroxide in methanol for 15 min, then heat-incubated with tris-EDTA buffer pH 9 at 96 °C for 21 min, followed by incubation with the primary MUM1 antibody at 1:50 dilution for one hour. Thereafter, the BioGenex Super Sensitive^TM^ Polymer-HRP IHC Detection System (BioGenex, Fremont, CA 94538, USA, code QD420-YIKE) was used according to the manufacturer’s instructions.

Sections were further incubated with a DAB chromogen (Dako, Glostrup, Denmark, code K3468) for 1–2 min, followed by routine counterstaining in Mayer’s haematoxylin, washing in running tap water, dehydration in alcohol and xylene, and coverslipping. Light microscopy examination of the IHC sections revealed the neoplastic cells were positively labelled for MUM1 (Figure 1d) compared to positive MUM1 control tissue of a cutaneous plasmacytoma from a 16-year-old dog (Figure 1d inset). The diagnosis was EMP.

A conservative treatment regime was administered to the bat at the rehabilitation centre, which included daily castor oil compresses to the mass and administration of vitamin C. However, the condition of the bat kept deteriorating, showing loss of body weight, poor appetite, and nausea, as well as isolating itself from the other bats. Two weeks later, the bat was anaesthetised, and the 1.5 × 1.0 cm lobulated mass was surgically excised (Figure 2a,b).

Unfortunately, the bat died under the anaesthetic. The excised mass and the entire bat were fixed with 10% formalin for processing and additional histopathological analysis. The mass showed the same neoplastic cell population as was observed in the biopsy (Figure 3a).

Examination of the other tissues showed moderate-to-severe locally extensive infiltration of the spleen (Figure 3b) as well as low numbers of neoplastic cells in circulation, including in the liver sinusoids (Figure 3c) and moderate numbers in the bone marrow (Figure 3d). The main differential diagnosis was multiple myeloma (MM) with skin involvement; however, the histological presentation, together with the largest mass being present in the skin, favoured a diagnosis of EMP with systemic metastases. In addition, another possible but less likely differential diagnosis considering the round cell population was lymphoma. This was excluded using standardised CD3 and CD20 immunohistochemistry for T-lymphocytes and B-lymphocytes, respectively (Figure 3e,f).

## 3. Discussion

Despite their relatively long life-spans, reports of neoplasia in *Chiroptera* species are rare. Indeed, there are only a handful of case reports of tumours found in bats [21,22,23,24,25]. Furthermore, a retrospective study looking at incidences of neoplasia within a captive population of male Egyptian fruit bats (*Rousettus aegyptiacus*) in 2004–2014 only found five cases [26], and an international collaboration examining bats from Australia, Asia, and Africa did not find any tumours during an analysis of a large number of individual bats from more than ten different species [27]. This apparent mitigation of tumourigenesis in bats has led to several groups investigating the potential mechanisms of tumour suppression in these animals. Some studies have proposed a critical role for the physiology of the mitochondria in bats, which appears to be more efficient and produces fewer reactive oxygen species (ROS) during metabolism than comparable non-flying mammals [28], and the mitigation of mitochondrial ROS at the site of production plays a key role in both avoiding tumourigenesis and extending longevity. Sequencing of the genome and transcriptome of the Brandt’s bat (*Myotis brandtii*), the longest-lived bat species known to-date, revealed unique sequence changes in the growth hormone receptor (GHR) [29]. This is of interest, as GHR mutations or growth hormone (GH) signalling deficiencies have been associated with increased resistance to cancer in humans and mice [30,31]; thus, reduced GH/GHR signalling may be a contributing factor to cancer-resistance in these long-lived bats [29].

A study that compared gene expression in the long-lived greater mouse-eared bats (*Myotis myotis*) to that seen in pigs, cows, and humans, found most of the microRNAs (miRNAs) identified were bat-specific and involved in regulating genes in tumourigenesis, ageing and immune pathways [32]. However, 37 orthologous miRNA groups shared across all four species were identified, of which six were differentially expressed, and in bats, three of the four upregulated miRNAs likely function as tumour suppressors against a variety of tumour types, while one of the down-regulated miRNAs is known to act as a promoter of tumourigenesis in human pancreatic and breast cancer [32]. Furthermore, cross-species analysis of the mRNAs also revealed bat-specific gene expression patterns, with 127 up-regulated genes being mostly enriched in DNA repair mechanisms and mitotic cell cycle activities, while 364 down-regulated genes were enriched for mitochondrial activity [32]. In a follow-on study, analysis of RNA from 150 blood samples from known aged bats found they did not show the same age-related transcriptomic changes commonly seen other mammals; instead, they showed a unique, age-related pattern of gene expression associated with autophagy, DNA repair, immunity, and tumour suppression, which may be responsible for their extended lifespans [33]. Recently, long-read sequencing of the genomes of two bat species (the Jamaican fruit bat (*Artibeus jamaicensis*) and the Mesoamerican mustached bat (*Pteronotus mesoamericanus*)) revealed the rapid evolution of both immune- and cancer-associated genes [34]. With six DNA-repair genes and 33 tumour suppressors showing signs of positive selection, it was suggested that this may contribute to the increased longevity and reduced cancer rates seen in bats [34]. Another study identified a role for the ATP-binding cassette (ABC) transporter ABCB1 in DNA damage resistance in bats and proposed that their ability to transport toxic chemicals out of their system, thereby protecting them from DNA damage induced by genotoxic compounds, was a potential mechanism of tumour suppression that contributes to their low incidence of cancer [35].

This report is the first recording of plasmacytoma in a bat, and as such, we compared the findings with plasmacytomas in dogs and cats. As was in this case report, most cutaneous EMPs are solitary; however, some animals present with multiple tumours [36]. They typically present as small, slightly raised dermal nodules, with skin showing loss of hair and occasionally ulceration [6]. As in humans, canine and feline EMPs have been subclassified based on histological features, including asynchronous, cleaved, hyaline, mature, and polymorphous subtypes [3,4,12,13]; however, they have not been associated with any prognostic significance [3]. The presence of amyloid can be a helpful diagnostic feature for plasmacytomas, and, although not observed in this case, it has been reported in ~10% of canine plasmacytomas [6] and one study of feline plasmacytomas found its presence in 3/9 (33%) cases [12]. A study of eight canine and two feline cases of EMP found that, immunohistochemically, the amyloid reacted positively with cross-reacting antibodies against human and equine Aλ amyloids [37]. A case report of a primary duodenal plasmacytoma with associated amyloidosis in a cat found that the cause of death, 96 days after initial diagnosis and despite surgical removal of the mass and post-operative chemotherapy, was a result of a complication of the tumour itself and associated amyloidosis [9]. However, in general, the prognostic significance of amyloids currently remains unclear, with some studies finding an association with a more aggressive behaviour and others not finding any clinical correlation [4,14,38].

The mitotic count of plasmacytomas are variable but usually low [6]. In this case, report, however, the mitotic count was high, and the presence of marked nuclear pleomorphism with occasional bi- and multinucleation was also observed, suggesting an aggressive tumour. Indeed, a report of a canine oral EMP that metastasised to both kidneys found the tumour cells showed aggressive features, including high nuclear pleomorphism, multinucleated cells, and a high number of mitotic figures [19]. Consistent with this, the EMP in this case showed evidence of systemic metastases, with neoplastic cells in circulation, specifically in the liver and bone marrow, as well as a locally extensive infiltration of the spleen.

MUM1, encoded by the *IRF4* gene, is a regulatory protein involved in the differentiation of B-lymphocytes to plasma cells and is required for immunoglobulin light-chain rearrangement. It is useful to identify plasmacytomas, MMs, and B-cell lymphomas in both canines and felines [6,14,19]. Indeed, of the B-cell markers commonly used in veterinary species, MUM1 is the preferred plasma cell marker for normal and neoplastic plasma cells, with one study reporting 84% of plasma cell tumours showing positive labelling, compared to 59/105 (56%) for CD79a and 21/108 (19%) for CD20 [39]. Furthermore, the study found that canine lymphomas that express MUM1 are few and typically of B-cell origin, and other canine leukocytic and melanocytic tumour do not express MUM1 [39]. The study concluded that mouse monoclonal antibody Mum-1p (which was used in this case report) is very specific for canine plasmacytomas and is superior in sensitivity and specificity to CD79a and CD20 for the identification of canine plasmacytomas in formalin-fixed, paraffin-embedded tissues [39]. Labelling is typically nuclear with a weak cytoplasmic component [6,39], which is consistent with that seen in this case report. This is the first reported use of MUM1 immunohistochemistry in a bat.

A differential diagnosis for EMP is MM, a multicentric plasma cell disease arising typically in the bone marrow. Indeed, in humans, although EMPs generally have a good prognosis, 11–36% of cases have been reported to proceed to MM [40,41,42]. Some studies in canines have found EMPs tend to be benign (although oral EMPs are locally aggressive), carrying an excellent prognosis following complete surgical excision [1,2,43,44], although a recent study found that up to a third of cases had a progression of plasma cell disease, with two cases showing myeloma-like progression [45], and there are occasional case reports of dogs with EMPs showing progression to a disseminated myeloma [46]. However, cutaneous involvement in patients with MM is very uncommon and usually occurs in the late stages of the disease, with the development of multiple skin lesions reflecting the increased tumour cell burden [47,48,49]. By contrast, this case had the skin mass as the largest lesion, with only locally extensive infiltration of the spleen and the presence of scattered neoplastic cells in the circulation, including into the liver sinusoids and bone marrow. Indeed, a study of cutaneous EMP in dogs found that the presence of intravascular neoplastic cells was both relatively frequent (20/125 (16%) cases) and did not correlate with local recurrence, clinically relevant metastases, or MM [50]. Thus taken together, as well as bearing in mind the lack of osteolytic bone lesions that are commonly seen in MM [51], this favours a diagnosis of EMP with systemic metastases. Another differential diagnosis for EMP is lymphoma. For example, a report of a cutaneous lymphoblastic lymphoma with systemic metastases in a cat found that, histopathologically, the tumour showed heterogeneous cell components, such as small lymphocytes, well-differentiated plasma cells, and plasmacytoid transformed lymphocytes [52]. By immunohistochemistry, the neoplastic well-differentiated plasma cells and plasmacytoid transformed lymphocytes were positive for MUM1 and λ-Ig light chains, and the neoplastic small lymphocytes were positive for CD20 [52]. By contrast, however, in this report, the lymphocytes scattered amongst the neoplastic cells were very few; they displayed normal lymphoid morphology, and some were positive for CD3 while others were positive for CD20 in more or less equal numbers, thus favouring a diagnosis of EMP with systemic metastases.

Thus, this case report adds to our limited knowledge of tumours in bats, describing the first report of a plasmacytoma in a bat, and also builds our appreciation for the multiple presentations of EMP in animals.

## Figures and Tables

**Figure 1 vetsci-11-00072-f001:**
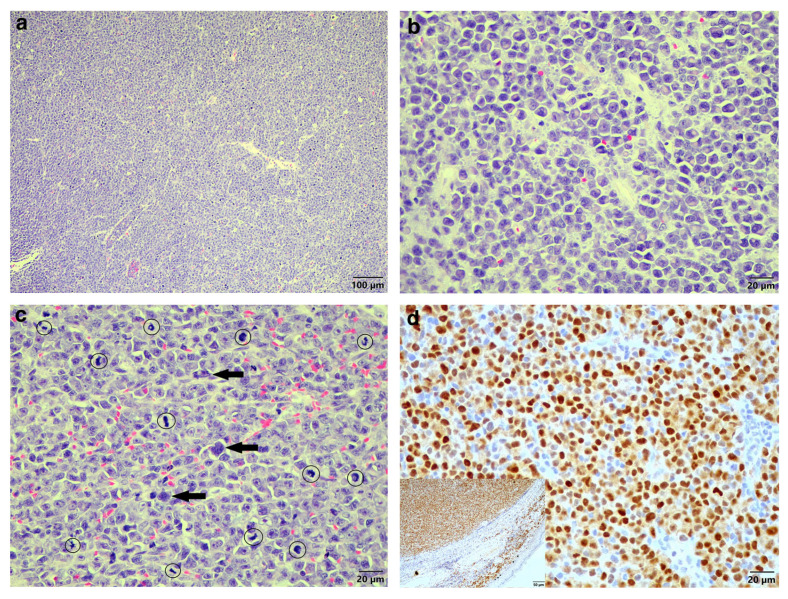
Histologic findings of the original biopsy from the cutaneous mass of the Cape serotine bat. (**a**) Sheets of neoplastic round cells (HE stain, 100× magnification). (**b**) The neoplastic plasma cells displaying a clear zone perinuclearly (HE stain, 400× magnification). (**c**) Marked nuclear pleomorphism, occasional multinucleation (arrows) and numerous mitoses (circles) (HE stain, 400× magnification. (**d**) MUM1-specific IHC labelling of the neoplastic cells (MUM1 IHC, 400× magnification). Inset: MUM1-specific positive control tissue with the epidermis and compressed dermal collagen serving as internal negative control (MUM1 IHC, 200× magnification).

**Figure 2 vetsci-11-00072-f002:**
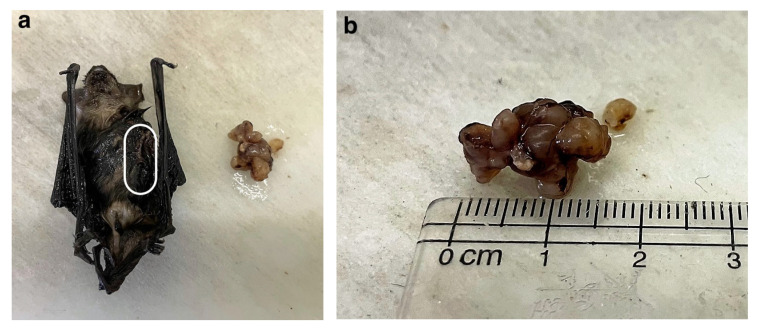
Macroscopic images of the Cape serotine bat and the plasma cell tumour. (**a**) Macroscopic image of the tumour and the bat after formalin fixation (frontal view with the white line demarcating where the tumour was located before excision). (**b**) Close-up image of the 1.5 × 1.0 cm lobulated tumour.

**Figure 3 vetsci-11-00072-f003:**
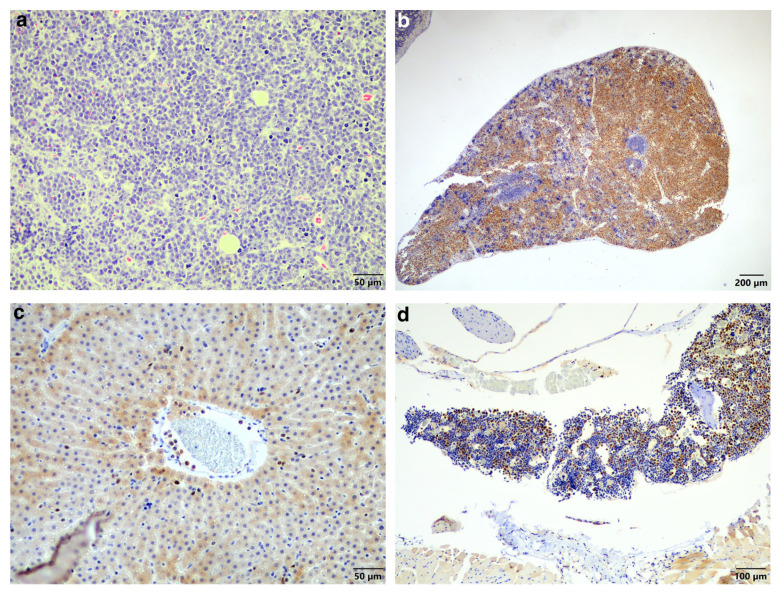
Histopathology of the plasma cell tumour, spleen, liver, and bone marrow. (**a**) Similar to the original biopsy, the mass revealed sheets of neoplastic round cells, occasionally forming packets (HE stain, 200× magnification). (**b**) Spleen showing locally extensive infiltration of MUM1-positive neoplastic cells (MUM1 IHC, 40× magnification). (**c**) Liver showing MUM1-positive neoplastic cells in circulation. Here, they are present in a small venule as well as in some sinusoids (MUM1 IHC, 200× magnification). (**d**) Bone marrow showing moderate infiltration by MUM1-positive neoplastic cells (MUM1 IHC, 100× magnification). (**e**) CD3-specific positive labelling normal T-lymphocytes scattered amongst the non-labelling neoplastic cells (CD3 IHC, 400× magnification). (**f**) CD20-specific positive labelling normal B-lymphocytes scattered amongst the non-labelling neoplastic cells (CD20 IHC, 400× magnification).

## Data Availability

All data underlying the results are available as part of the article, and no additional source data are required.

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
