# Peer review of "Cutaneous Plasmacytoma with Systemic Metastases in a Cape Serotine Bat (Laephotis capensis)"

_vetsci, 2024, doi:10.3390/vetsci11020072_

Round 1
Reviewer 1 Report
Comments and Suggestions for Authors
The paper focuses on an important topic and adds relevant information in diagnostic pathology. Although the morphological evaluation associated with immunopositivity for MUM1 are sufficient for the diagnosis, I suggest the inclusion of the cd79 and cd20 markers considering a comparative pathology approach.
Author Response
Reviewer 1:
The paper focuses on an important topic and adds relevant information in diagnostic pathology.
Thank you, we appreciate your positive feedback.
Although the morphological evaluation associated with immunopositivity for MUM1 are sufficient for the diagnosis, I suggest the inclusion of the cd79 and cd20 markers considering a comparative pathology approach.
From experience, we have concluded that in general, CD20 is a more reliable marker to distinguish neoplastic B-lymphocytes from neoplastic plasma cells compared to CD79 and since this is a diagnostic case report, we have decided to perform CD20 and CD3 in order to exclude a lymphoid neoplasm such as lymphoma in this case. A follow-up study evaluating the immunoreactivity of MUM1, CD79 and CD20 to establish the overlap of labelling of the different cell types originating from B-cell precursors in a comparative fashion for bats specifically may be of value, but that is beyond the scope of this diagnostic case report.
Reviewer 2 Report
Comments and Suggestions for Authors
This case report is a very well written description of a case of plasmacytoma with systemic metastasis in a Cape serotine bat. Given the generally rare frequency of cancer in bats which is of interest in the cancer field, and that this manuscript represents the first report of plasmacytoma in bats, the report is worthy of publication. Minor comments/concerns listed below should be addressed before consideration for publication.
Line 132-133: “mild presence” is a vague description. Did you intend to communicate that neoplastic cells were observed in low frequency in the circulation? If so, the mention of frequency of neoplastic cells in the circulation in terms of low, moderate, or high would be a more appropriate description for the different sites (circulation, spleen, liver, and bone marrow). Based on Figure 3, the frequency of tumor cells appeared relatively high in the spleen and bone marrow.
Line 156: You might consider mentioning findings of other references related to bats’ resistance to cancer such as: Genome Biol. Evol. 15(9) https://doi.org/10.1093/gbe/evad148 and Nature Ecology & Evolution, Vol 3, July 2019, 1110–1120.
In the description of the methods for IHC analysis for MUM1 staining, a brief mention of a staining control should be included. Additionally one slide demonstrating a negative control should be included in Figure 1.
Author Response
Reviewer 2:
This case report is a very well written description of a case of plasmacytoma with systemic metastasis in a Cape serotine bat. Given the generally rare frequency of cancer in bats which is of interest in the cancer field, and that this manuscript represents the first report of plasmacytoma in bats, the report is worthy of publication.
Thank you, we appreciate your positive feedback.
Minor comments/concerns listed below should be addressed before consideration for publication.
Line 132-133: “mild presence” is a vague description. Did you intend to communicate that neoplastic cells were observed in low frequency in the circulation? If so, the mention of frequency of neoplastic cells in the circulation in terms of low, moderate, or high would be a more appropriate description for the different sites (circulation, spleen, liver, and bone marrow).
Thank you, we agree that “mild presence” is vague, we have changed this to “Examination of the other tissues showed moderate to severe locally extensive infiltration of the spleen (Figure 3b) as well as low numbers of neoplastic cells in circulation, including in the liver sinusoids (Figure 3c) and moderate numbers in the bone marrow (Figure 3d).” Please see lines 153-161.
Based on Figure 3, the frequency of tumor cells appeared relatively high in the spleen and bone marrow.
Sorry for the confusion, considering the overall presence throughout all the organs it was low, but as you pointed out the spleen and bone marrow contains more than just low numbers. We have changed it accordingly in line 153-161 as mentioned above.
Line 156: You might consider mentioning findings of other references related to bats’ resistance to cancer such as: Genome Biol. Evol. 15(9) https://doi.org/10.1093/gbe/evad148 and Nature Ecology & Evolution, Vol 3, July 2019, 1110–1120.
Thank you for this suggestion, we have added these references as well as some additional ones, and expanded the section on putative mechanisms for cancer resistance in bats. Please see lines 170-207.
In the description of the methods for IHC analysis for MUM1 staining, a brief mention of a staining control should be included. Additionally one slide demonstrating a negative control should be included in Figure 1.
We have identified the positive control tissue as suggested, please see line 119-120, and included an image (figure 1d inset) where the epidermis and compressed pre-existing dermal collagen serves as the negative internal control compared to the neoplastic cells of the positive control mentioned in line 112-113. Thank you for the suggestion. We have also replaced figure 1d with an image with better white balance.
Reviewer 3 Report
Comments and Suggestions for Authors
Dear Authors, the work is very interesting and implements knowledge on the topic. Furthermore, it stimulates the scientific community to express interest in this animal species.
From my standpoint, this case report is original and describes the first case of tumor in a Cape serotine bat; furthermore, represents the first case of round cell tumor found in this species. Given the rarity of the case, the work adds important information in this thematic area.
Elements that the AA can be improved, although the HE images are suggestive of plasma cell tumor - MUM positive, would be to investigate the cellular phenotype using markers for lymphocytes, excluding lymphoma in the differential diagnosis. Furthermore, the figure 3a could have be taken at a higher magnification for greater morphological detail.
Author Response
Reviewer 3:
Dear Authors, the work is very interesting and implements knowledge on the topic. Furthermore, it stimulates the scientific community to express interest in this animal species.
Thank you, we appreciate your positive feedback.
From my standpoint, this case report is original and describes the first case of tumor in a Cape serotine bat; furthermore, represents the first case of round cell tumor found in this species. Given the rarity of the case, the work adds important information in this thematic area.
Thank you.
Elements that the AA can be improved, although the HE images are suggestive of plasma cell tumor - MUM positive, would be to investigate the cellular phenotype using markers for lymphocytes, excluding lymphoma in the differential diagnosis.
Thank you for this suggestion. We have performed CD3 and CD20 immunohistochemistry on the tumour to rule out the possibility of a lymphoma. Please see lines 160-161 and figures 3e and 3f with their legends.
Furthermore, the figure 3a could have be taken at a higher magnification for greater morphological detail.
We agree and we have made this change. Please see the updated figure 3a.